# Comparative expression profiling reveals widespread coordinated evolution of gene expression across eukaryotes

Trevor Martin[1] & Hunter B. Fraser[1]

Comparative studies of gene expression across species have revealed many important insights, but have also been limited by the number of species represented. Here we develop an approach to identify orthologs between highly diverged transcriptome assemblies, and apply this to 657 RNA-seq gene expression profiles from 309 diverse unicellular eukaryotes. We analyzed the resulting data for coevolutionary patterns, and identify several hundred protein complexes and pathways whose expression levels have evolved in a coordinated fashion across the trillions of generations separating these species, including many gene sets with little or no within-species co-expression across environmental or genetic perturbations. We also detect examples of adaptive evolution, for example of tRNA ligase levels to match genome-wide codon usage. In sum, we find that comparative studies from extremely diverse organisms can reveal new insights into the evolution of gene expression, including coordinated evolution of some of the most conserved protein complexes in eukaryotes.

[1] Department of Biology, Stanford University, Stanford, CA 94305, USA. Correspondence and requests for materials should be addressed to H.B.F. (email: hbfraser@stanford.edu)

Many cellular functions are carried out by groups of proteins that must work together, such as pathways and protein complexes. When one of these functions is no longer needed by a particular species, then there is no longer any selection to maintain the genes needed specifically for this function, and they will eventually deteriorate into pseudogenes or be lost altogether. A method known as phylogenetic profiling (PP) leverages this idea, correlating patterns of gene presence/absence across species to identify functionally related genes[1]. For example, this technique has been used to discover novel genes involved in Bardet–Biedl syndrome[2–4] and mitochondrial disease[5], since these diseases involve genes that have been lost in multiple independent lineages. In these studies, patterns of gene conservation across species are typically represented by their binary presence/absence, and knowledge of the species phylogeny is used to identify genes whose losses have coincided with those of well-characterized genes[6]. Coordinated gene losses can then be analyzed for gene pairs individually or gene groups as a whole to reveal functional relationships[7].

In addition to the correlated gene losses that are the focus of PP, functional similarity is also suggested by conserved coexpression, where gene pairs are coexpressed across various environmental conditions in multiple species[8–10]. A complementary approach is to search for correlated expression across species, rather than across environments within species. For example, coordinated evolutionary changes have been observed between computationally predicted expression levels (based on codon usage bias) in yeast and other microbes[11,12]. Experimentally measured gene expression levels could also potentially uncover genes with correlated evolution, including genes that are never lost and thus not amenable to PP; however, in practice this has not been possible because of the small number of species, and the narrow phylogenetic breadth, in previous studies of gene expression evolution. The largest such studies have been limited to a few dozen species and have focused exclusively on mammals[13,14] or yeast[15], in contrast to recent PP studies that utilize hundreds of complete genome sequences from widely divergent species[6,16,17].

The Marine Microbial Eukaryotic Transcriptome Project (MMETSP)[18] recently generated what is by far the largest multispecies gene expression data set to date, both in terms of the number of species and the phylogenetic diversity, with RNA-seq for 657 samples from 309 species. These species are eukaryotic marine microbes collected from across the world, spanning most major eukaryotic lineages, including many rarely studied phyla that lack even a single sequenced genome (Fig. 1a)[18]. To put this diversity in perspective, most pairs of MMETSP species are even more diverged than fungi and animals. All RNA samples were prepared, sequenced, and analyzed following a standardized pipeline established by the MMETSP. Some of these data have been examined in studies of specific species[19–22], but the data have not previously been analyzed collectively.

In this work we identified ortholog groups and assembled the hundreds of transcriptomes into a single matrix, which we hope will be a valuable resource for future studies of gene-expression evolution. We illustrate two applications made possible by these data: identifying coordinated evolution of gene expression in protein complexes, and detecting gene expression adaptations to both intra- and extracellular environments. We have also made the full data set available in an interactive website at http://mmetspdata.appspot.com.

## Results

### Creating a single expression matrix for comparative study.
A major challenge in asking evolutionary questions with the MMETSP data is that orthologous genes must first be identified across hundreds of species. Although many databases of orthologs exist, these cannot be easily applied to the de novo assembled transcriptomes of the MMETSP. Therefore, we developed an approach to identify orthologous groups, allowing us to create a single gene expression matrix to facilitate large-scale comparative studies.

To identify orthologous groups, we utilized a two-step approach: first matching each transcript to a group of related proteins (represented by UniProt100 IDs, which are nonredundant clusters of protein sequences; see Methods), and then merging redundant hits in an iterative process (Fig. 1b). The step of merging had several complexities. For example, if a transcript's top UniProt100 match was only observed once across all RNA-seq samples, then it was deemed uninformative for comparative studies, and the next best match (ranked by BLASTP alignment scores) was then tested. Up to five UniProt100 IDs were tested for each transcript. In addition, if a single sample had multiple matches to the same UniProt100 ID—potentially representing transcript isoforms or paralogs—these were merged and their read counts were summed within each sample, to reflect the total expression level of this gene or gene family in each sample. For further details, see the Methods and Supplementary Figure 1.

This approach resulted in a single unified expression matrix of 4219 genes by 657 samples, where every gene had detectable expression in at least 100 samples (Fig. 1c; Supplementary Table 1). This matrix is available for download or interactive search at http://mmetspdata.appspot.com. For example, users can search by gene symbol, and visualize the relationship between expression of a gene across all samples compared to variables such as each sample's latitude, longitude, depth, temperature, salinity, or pH.

### Detecting coordinated evolution of gene sets.
Having generated the unified expression matrix, we next turned to the question of whether we could identify signals of coordinated evolution. We term this approach—which focuses on coexpression between species, rather than within species[8–10]—"phylogenetic expression profiling" (PEP). This is conceptually similar to PP (Fig. 2a), but it does not require that genes have been lost from any lineages, and thus is applicable to any gene whose expression can be measured across many species.

We first calculated all pairwise Spearman correlations between genes in the expression matrix. Because of the complex phylogenetic structure of the data, which can inflate correlations due to non-independence, we did not attempt to assign $p$ values to individual pairwise PEP correlations; rather we focused on detecting coordinately evolving groups of genes, for which we can create a random permutation-based null distribution that precisely captures the effects of phylogenetic structure, even when the phylogeny is not known[23]. In brief, we selected random sets of genes which may have correlated expression due to phylogeny, but should not have any effect of shared function, thus providing an estimate of PEP correlations expected due to phylogenic structure (see Methods). Any gene set whose PEP correlation exceeds this expected distribution was deemed significant after correction for phylogenetic structure. For the purposes of comparing our results to PP, we restricted our analysis to samples in which a given pair of genes were both detectably expressed, so that PEP does not utilize gene presence/absence information that is the basis of PP. However, for other applications the samples with zero expression of a gene could be included to add more information for many gene sets.

To test the performance of PEP, we compared our results to PP in two ways. First, we examined genes with a known role in cilia,

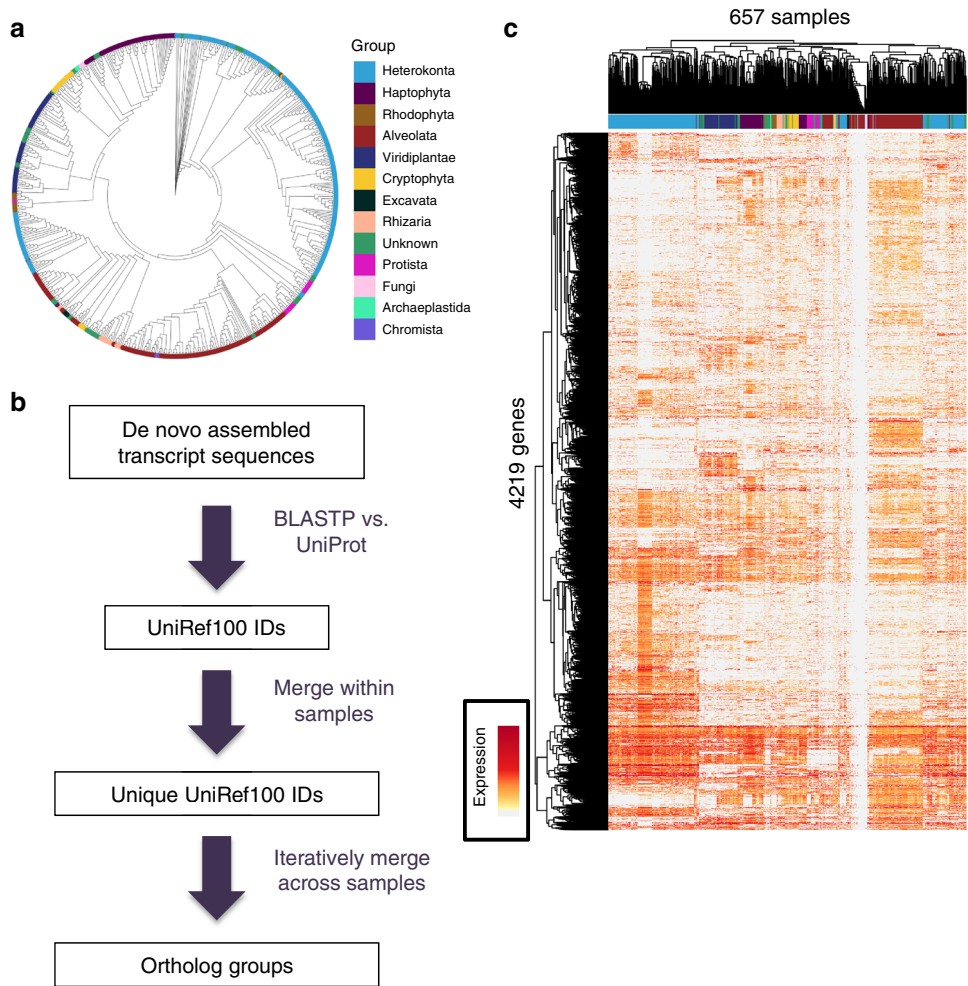

**Fig. 1** Overview of phylogenetic expression profiling approach and data. **a** 18S-based cladogram of the species in this study. **b** Overview of our approach for inferring ortholog groups from RNA-seq transcripts (see also Methods and Supplementary Figure 1). **c** Heatmap of expression levels of the 4219 genes and 657 samples analyzed in this study. Samples and genes are clustered hierarchically. Color bar near the top shows the phylum of each sample, matching the colors in **a**

since this organelle is one of the most significant gene sets implicated by many PP studies[6,16,24]. We found this gene set was also enriched for high PEP correlations (Supplementary Figure 2; permutation $p = 5.1 \times 10^{-5}$), indicating that the ciliary genes show coordinated evolution of gene expression, in addition to gene loss. We then compared the two methods at a finer scale, by asking whether the specific ciliary gene pairs with the strongest PP signal also show coordinated evolution by PEP. Comparing the PEP correlations to the binary presence/absence PP correlations, we found a high level of agreement (Fig. 2b; permutation $p = 3.4 \times 10^{-22}$), suggesting that the specific ciliary gene pairs most likely to be lost together also tend to have coordinately evolving expression levels.

We then asked whether PEP and PP agree at a more broad scale, by testing whether the 327 coordinately evolving modules identified in a recent PP study[6] showed increased PEP signal as well. We found significant (permutation $p = 2.9 \times 10^{-47}$; see Methods) enrichment for PEP correlations in these previously identified PP modules (despite our exclusion of samples with no detectable expression of a given gene), suggesting that PEP detects many of the same gene sets implicated by PP. For example, some of the strongest PEP correlations were among genes involved in the ribosome, spliceosome, and cilia.

To identify additional coordinately evolving modules not detected by PP, we applied PEP to a collection of 5914 previously characterized gene sets, including both pathway and disease databases (see Methods). Of these, we found 662 gene sets with significant coordinated evolution, compared to only ~33 expected at this level by chance (Fig. 2c; 5% false-discovery rate (FDR); Supplementary Table 2). Most of these had no previous evidence of coordinated evolution from PP studies, such as RNA degradation, the proteasome, and the nuclear pore complex. Examining all pairwise PEP correlations within each of these gene sets revealed that the coordinated evolution was shared across most gene pairs, rather than only driven by a small subset of them (e.g., as shown for proteasome genes in Fig. 2d). Many of these coordinately evolving gene sets have not been detected by PP because of the extreme rarity of losing these genes (Supplementary Figure 3).

We also compared these coordinately evolving gene sets from PEP to coexpression analysis within a single species, the yeast *Saccharomyces cerevisiae*, across environmental or genetic perturbations. Using a compendium of expression profiles measured in 121 environments[25] we calculated the median coexpression correlations for each pair of genes within gene sets that were found as significant using PEP and compared the two (Fig. 3a). There was an overall weak correlation between the two sets (Pearson $r = -0.08$, $p = 0.14$), suggesting that the PEP method and coexpression across environments are largely orthogonal. Repeating this analysis for genetic

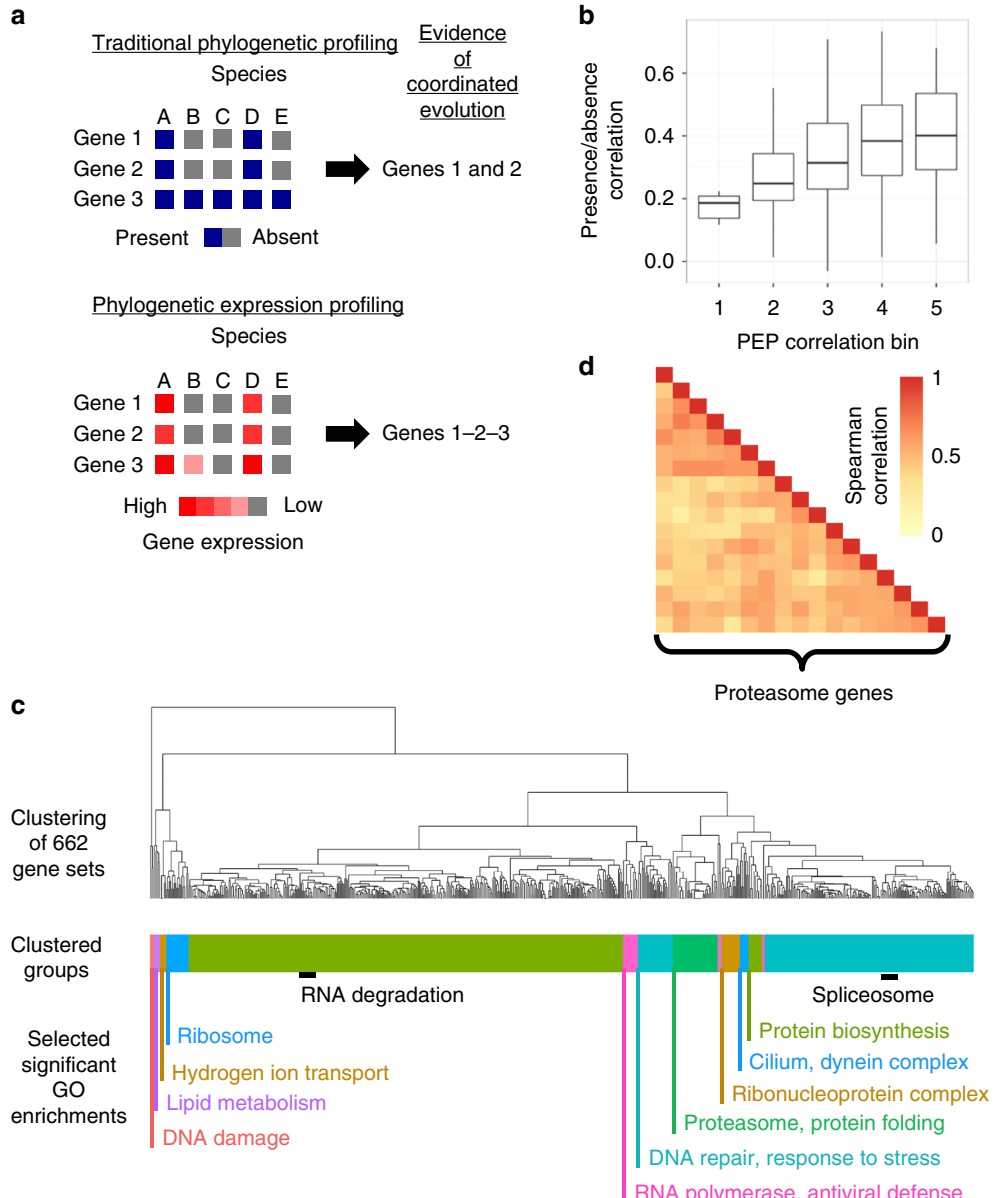

**Fig. 2** Phylogenetic expression profiling reveals coordinated evolution within gene sets. **a** Overview of traditional phylogenetic profiling (PP; top) and phylogenetic expression profiling (PEP; bottom). PEP uses the quantitative gene expression levels across species rather than the binary presence/absence of a gene. Patterns of coordinated evolution hidden to PP can be potentially uncovered using PEP. **b** For ciliary genes, pairwise PP correlations increase with PEP correlation strength. **c** The 662 gene sets with significant PEP scores are clustered by the pairwise correlations between gene sets. The color bar below the dendrogram shows the 15 unique gene set groups the dendrogram was divided into and gene ontology enrichments for each group are highlighted in the same color. Black bars highlight notable groups of gene sets within the larger groups. **d** Proteasome genes were found to be undergoing coordinated evolution and are shown as a heatmap

perturbations—coexpression across 1484 yeast gene deletion strains[26] (Fig. 3b)—we observed a similarly low correspondence (Pearson $r = 0.03$, $p = 0.62$). From the environmental yeast data, we identified the deadenylation dependent mRNA decay gene set as having evidence for coordinated evolution by PEP but not for coexpression across environments (Fig. 3c). Conversely, the ribosome gene set was significant in both PEP and in the cross-environment analysis, but had a stronger signal in the environmental analysis (Fig. 3d).

We then asked whether we could also detect coordinated evolution between, rather than within, gene sets. To identify these, we calculated the PEP correlation between each pair of genes in a given pair of gene sets (excluding any genes present in both; see Methods). Among the 218,791 pairs of gene sets we

tested, 22,665 had evidence of coordinated evolution (with <1 expected by chance; Supplementary Table 3). For example, we found that genes involved in the Golgi apparatus had strong evidence (permutation $p = 2.9 \times 10^{-5}$) of coordinated evolution with genes downregulated in Alzheimer's disease (Supplementary Figure 4A). Previous studies have implicated Golgi fragmentation in the pathogenesis of Alzheimer's[27,28] and this coordinated evolution suggests that these gene sets may be functionally associated even in microbes.

In addition to identifying coordinated evolution within and between known gene sets, PEP can also implicate novel genes evolving in tandem with a known gene set. For this analysis, we calculated the PEP correlation between the genes in a given set and every other gene; those with the strongest median

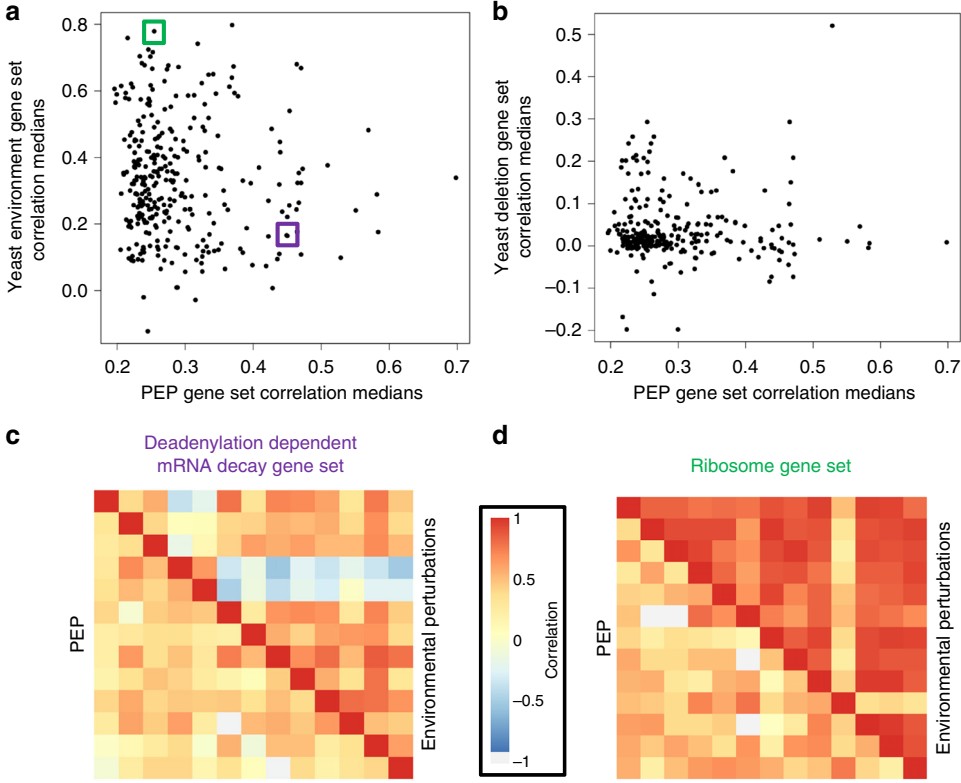

**Fig. 3** Comparison of PEP with cross-environment and cross-genotype expression profiling methods. **a** Scatterplot of the gene-set correlation medians for the gene sets found as significant using the PEP method introduced here vs. the correlation medians for the same gene sets in a data set of yeast expression across environments. The gene sets boxed in purple and green are highlighted in **c** and **d**, respectively. **b** Scatterplot of gene-set correlation medians as in **a**, but comparing PEP to yeast expression across deletion strains (i.e., across genotypes). **c** Comparison of gene-gene correlations for the deadenylation dependent mRNA decay gene set highlighted in **a** for the PEP method (lower triangle) and classical yeast coexpression correlations across environments (upper triangle). **d** Comparison of gene–gene correlations for the ribosome gene set highlighted in **a** as described in **c**

correlations are most likely to be functionally related to that set. For example, *TULP2*—a member of the tubby-like gene family—had the highest PEP correlation with the diabetes pathway gene set (Supplementary Figure 4B; permutation $p = 3.0 \times 10^{-4}$), and it is also a candidate gene for severe obesity, a closely related trait[29]. More broadly, genes coordinately evolving with diabetes pathways were enriched for type 2 diabetes GWAS associations (permutation $p = 2.3 \times 10^{-2}$ for the top 10 genes, and permutation $p = 2.6 \times 10^{-2}$ for the genome-wide trend; see Methods), suggesting that the PEP correlations may be at least somewhat predictive of genes involved in T2D.

**Identifying potential gene-expression adaptations**. Correlations between genotypes or phenotypes and environmental variables can potentially indicate local adaptation to the environment, though this approach is usually applied either within species or between closely related species[35,36]. We sought to determine if the same strategy could be applied to our diverse set of microbes, to generate candidate examples of adaptive evolution.

Latitude is a major driver of local adaptation, resulting in trends such as smaller animals close to the equator (Bergmann's Rule)[30]. In addition, latitudinal gradients in skin pigmentation and gene expression have been observed in recent human evolution[23,31]. To investigate whether any gene expression levels show a latitudinal gradient across the diverse set of MMETSP species, we correlated absolute latitude (where each sample was collected) with expression levels of every gene. Although we did not find any functions enriched in the latitude-associated genes, some genes showed significant associations; the most strongly correlated gene was the translesion DNA polymerase *POLH*

(Fig. 4a; Spearman's $\rho = -0.52$, $p = 2.4 \times 10^{-24}$), which is required for accurate replication of ultraviolet (UV)-damaged DNA. Expression was generally higher closer to the equator, as expected if its mRNA level has evolved in response to the local levels of UV radiation. Indeed, average UV radiation was also predictive of POLH expression across MMETSP samples (Supplementary Figure 5; Spearman's $\rho = 0.47$, $p = 2.4 \times 10^{-7}$).

Other characteristics that might drive adaptations of each species can be estimated directly from the assembled transcriptomes. For example, in each species we calculated the genome-wide fraction of codons encoding each amino acid, and tested whether these fractions predict the expression levels of the corresponding tRNA ligases—enzymes that "charge" tRNAs with the appropriate amino acid. Of the ten tRNA ligases with expression data, all ten had a higher than median correlation with the relative abundances of their respective codons (binomial $p = 9.8 \times 10^{-4}$). For example, the association between the expression of the aspartate-tRNA ligase (*DARS*) and aspartate codon abundance is shown in Fig. 4b (Spearman's $\rho = 0.40$, $p = 4.8 \times 10^{-15}$). This supports our hypothesis of adaptive matching between tRNA ligase levels and genome-wide codon usage.

**Discussion**

Over the past 20 years, the rapid proliferation of genome sequences and genome-wide gene expression data from across the tree of life has led to a plethora of methods aimed at extracting information about genes from their evolutionary patterns. We hope that by assembling the largest comparative gene-expression data set to date in a convenient format, other researchers might

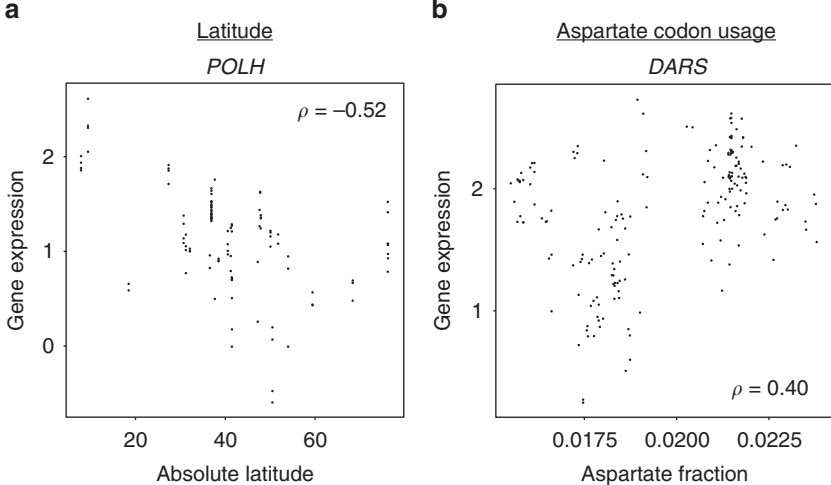

**Fig. 4** Associations between expression and other sample information. **a** Scatterplot of the association between gene expression and the absolute value of latitude for the DNA polymerase *POLH*. Each point represents one RNA-seq sample. **b** Scatterplot of the association between expression of the aspartate-tRNA ligase *DARS* and the abundance of aspartate codons in the coding regions of each sample's transcriptome

use these data to answer a wide range of questions. We provided two examples of topics that can be explored with these data: coordinated evolution of gene expression, and adaptive evolution of gene expression.

The PEP method builds on traditional PP as well as comparative gene-expression studies[11,12], and together with the most phylogenetically diverse gene-expression data set available to date, it revealed widespread evidence for coordinated evolution of gene expression. This is related but distinct from conserved coexpression within species[8–10] since such conservation could result from a lack of change in expression, whereas changes across many species are required for PEP to detect coordinated evolution. Interestingly, PEP—which in our initial implementation only relies on species where a gene is present—identified many gene sets previously implicated by PP, in addition to many novel sets of coordinately evolving genes. One explanation for why gene sets such as mismatch repair have not been identified by PP is that PP is substantially underpowered to detect these gene sets because of the rarity of their loss.

Further, our analysis of coordinated evolution between gene sets allowed us to infer potential functional linkages between known biological pathways. In particular, Golgi fragmentation in Alzheimer's disease has been linked to promotion of amyloid beta production[27] and potential phosphorylation of the tau protein, which underlies the formation of neurofibrillary tangles[32]. Since the Golgi genes are not themselves misregulated in Alzheimer's disease[33], a differential expression analysis of the Alzheimer's patient samples vs. controls could not provide this connection.

Previously, within single species or genera, many latitudinal gradients of traits have been reported, which are often attributed to local adaptations to climate[23,34–36]. This study has expanded such analyses to a much larger phylogenetic breadth. *POLH* is a strong candidate gene for local adaptation to UV, considering its role in the repair of UV-induced damage which leads to xeroderma pigmentosum when mutated in humans[37]. One caveat of this analysis is that it relies on the latitude of isolation of each strain and does not represent the breadth of latitudes where a species lives or the culture conditions. Other methods, such as directly measuring gene expression from environmental samples without culturing, could further elucidate these types of relationships. Additionally, our identification of tRNA ligase levels as associating with codon abundance suggests that tRNA ligase levels may adaptively evolve in response to species-specific codon

usage, and is consistent with patterns of tRNA gene copy number and codon usage in bacteria[38–40].

Overall, applying our PEP framework to a gene expression data set of unprecedented phylogenetic diversity, we identified many novel examples of coordinated evolution. These included thousands of cases of coordinated evolution within and between gene sets, and also coordinated evolution implicating novel genes related to diabetes pathways. Although it may at first seem surprising that unicellular eukaryotes could shed light on complex diseases like Alzheimer's and T2D, the fact that the genes are present throughout eukaryotes suggests that the underlying cellular functions are far more conserved than the specific human disease phenotypes—consistent with previous work, for example using yeast to study Parkinson's disease[41] and plants to study neural crest defects[42].

We expect that as the diversity of species with publicly available gene expression data continues to grow, PEP will become a powerful approach for detecting coordinated evolution at the molecular level, and for leveraging these patterns to inform us about functional connections between genes conserved throughout the tree of life.

## Methods
**Data filtering and normalization**. Raw reads and transcriptome assembled coding sequence (CDS) data for 669 individually annotated samples and 119 jointly annotated sample sets from the MMETSP were downloaded from the CAMERA database (http://camera.crbs.ucsd.edu/mmetsp/index.php). Details on each annotation method (performed by the MMETSP project) can be found on the MMETSP website (http://marinemicroeukaryotes.org/resources). All raw reads were then normalized using the Transcripts per Million (TPM) normalization technique[43]. Up to five Swissprot ID annotations provided by MMETSP for each CDS were then culled for IDs that had a BLASTP alignment score of at least 80% the maximum alignment score for that CDS (based on the observation that typically there is a steep dropoff of alignment scores after roughly this point), and converted to UniRef100 IDs (UniRef IDs are comprehensive nonredundant clusters of UniProt sequences[44]). In order to create vectors of expression across samples for a set of UniRef100 defined "genes", normalized read counts were combined within a sample for CDSs that had at least three matching UniRef100 IDs and then across samples by ranking the UniRef100 IDs by alignment score and then creating an expression vector for an annotation by matching the unique top ranked annotations against the top ranked annotation for each CDS in each sample with ties resolved by annotation score. Iterative matching is required as some CDSs do not share top ranked annotations, but do share lower ranked annotations. This initial round of matching was then followed by matching successively lower ranked annotations of still unmatched normalized read counts until all read counts are combined into expression vectors. Note that only unmatched read counts are carried over into each new iteration. For details see Supplementary Figure 1. The

expression vectors with at least 100 samples with measured expression were then combined into a matrix with each column as a sample (669 and 387 samples, for the individual and jointly annotated sets respectively) and each row as a gene (4995 and 1051 genes).

For the individually annotated samples, this expression matrix was then normalized by dividing each sample by the total number of genes in that sample and then adjusting for batch effects by regressing out the MMETSP transcriptome pipeline used (the two pipelines used differed in the method for transcriptome assembly), the day the sample was processed, and the lab that submitted the sample, setting the value of samples missing expression to zero for the regression step. All variables were regressed out as binary factors. Any samples missing any of these variables were dropped from the analysis.

For both sample annotations, the UniRef100 ID for each gene was converted to a UniRef50 ID (a more lenient across-species gene clustering than the UniRef100 ID) and any expression vectors with the same ID were collapsed by sum. The resulting individual annotation matrix has 4219 genes and 657 samples and the combined annotation matrix has 1031 genes and 387 samples.

**Phylogenetic-expression profiling.** Gene sets from the Online Mendelian Inheritance in Man (OMIM) database[45], the Human Phenotype Ontology (HPO) database[46], the Mouse Genome Informatics (MGI) database[47], and the Molecular Signatures Database (MSigDB)[48] were downloaded to create a list of 5914 gene sets with at least three genes that mapped to UniRef50 IDs in the individual annotation data set.

PEP tests for coordinated evolution of gene expression levels by calculating the median Spearman correlation between all pairwise combinations of genes in a gene set. Importantly, each pairwise correlation was calculated using only the samples that had expression measured for each gene and the genes that had at least 20 such samples. To calculate the significance of this median correlation, it was compared to 10,000 null median correlations created by random gene sets with the same number of genes, drawn from the 25 genes that most closely match the data missingness profile of the gene they replace. The data missingness profile for a gene pair was quantified by the Euclidean distance between the presence/absence vector of each gene across samples. The significance was then given by:

$$p \text{ value} = \frac{\left( \sum_{i=1}^{10,000} \phi_{\rho_i} + 1 \right)}{(10,001)}. \quad (1)$$

$$\phi_{\rho_i} \equiv \begin{cases} 1, \rho_i \geq \rho_{\text{obs}} \\ 0, \rho_i \leq \rho_{\text{obs}} \end{cases}. \quad (2)$$

The FDR is then determined by treating each of the 10,000 permutations as the real data and calculating 10,000 sets of $p$ values as above. A sliding $p$ value cutoff is then instituted and the ratio of $p$ values below this cutoff in the real data to the mean of the number of $p$ values below this cutoff in the 10,000 null permutations is the FDR.

**Comparison with PP.** Evolutionarily conserved modules (ECMs) from the clustering by inferred models of evolution (CLIME) algorithm applied to human pathways were downloaded from the CLIME website (http://www.gene-clime.org/). The 327 ECMs with an ECM score of greater than five and at least two genes in the individual annotation matrix were used for the validation test. To validate the PEP method, we calculated the median correlations for these ECMs in the same way as PEP, and the median of this distribution across ECMs was then compared to 10,000 null medians calculated using the same null strategy as PEP. The permuation $p$ value for enrichment for high PEP scores is then:

$$p \text{ value} = \frac{\left( \sum_{i=1}^{10,000} \phi_{\text{median}\rho_i} + 1 \right)}{(10,001)}. \quad (3)$$

$$\phi_{\text{median}\rho_i} \equiv \begin{cases} 1, \text{median } \rho_i \geq \text{median } \rho_{\text{obs}} \\ 0, \text{median } \rho_i \leq \text{median } \rho_{\text{obs}} \end{cases}. \quad (4)$$

Since the observed statistic was more extreme than all 10,000 permutations, a $z$-score based $p$ value was estimated:

$$z - \text{score} = \frac{\left( E\left[10,000 \text{ median } \rho_{\text{perm}}\right] - \text{median } \rho_{\text{obs}} \right)}{\left( \text{SD}\left[10,000 \text{ median } \rho_{\text{perm}}\right] \right)}. \quad (5)$$

$$p \text{ value} = \frac{2}{\sqrt{2\pi}} \int_{-\infty}^{-|z-\text{score}|} e^{-x^2/2} dx. \quad (6)$$

The PP correlation of a gene set was calculated by taking the median of the Pearson correlation of each pairwise presence/absence vector for each gene in the set. For a gene set, the PP correlation was then compared to the PEP correlation for

each gene by calculating the Pearson correlation between the PEP and PP correlations for each gene pair. The significance of this correlation was then calculated by permuting the presence and absence vectors for each gene in the set and then recalculating the PEP vs. PP correlation 10,000 times; the number of times a permutation beat or matched the observed value divided by the number of permutations was then the permutation $p$ value which was then converted to a $z$-score based $p$ value as above.

**Phylogenetic tree construction.** The 18S sequence available for 655 samples was downloaded from the CAMERA database as above and aligned using the multiple sequence alignment tool Clustal Omega[49]. This alignment was then used to create a maximum likelihood based tree using the program RaxML[50] with parameters: $-fa$ $-x$ 12345 $-p$ 12345 $-\#$ 100 $-m$ GTRGAMMA. 18S sequences that did not have available sample meta data were then dropped, leaving a total of 635 samples.

**Comparison with yeast coexpression.** Preprocessed expression data for 6423 *Saccharomyces cerevisiae* genes for yeast grown across 121 different conditions was downloaded from Caudy et al.[25], and yeast expression data for 6170 *S. cerevisiae* genes across 1484 deletion strains was downloaded from Kemmeren et al.[26]. Yeast gene IDs were matched to human gene names using Biomart and then mapped to the same gene sets used for the PEP analysis described above[51]. For each gene set, yeast median gene set correlations were calculated by taking the correlation for all pairwise combination of genes within that set, either across environments or across deletion strains.

**Gene-set pairwise comparison.** The correlation score between two gene sets was calculated by taking the median of the pairwise gene PEP correlations, excluding any genes present in both gene sets. A dendrogram relating the gene sets with significant PEP scores at a 5% empirical FDR as calculated above was created by calculating the matrix of correlation scores between all the significant gene sets, taking the Euclidean distance between the rows of this matrix, and then hierarchically clustering these distances using the complete linkage algorithm in R's hclust function[52]. Significance of individual gene set pairwise comparisons was calculated in two steps by first computing the $p$ value for the observed correlation between each of the gene sets in the comparison and 10,000 gene sets matched by phylogenetic profile and size as in the PEP method above. The maximum of these two $p$ values for random gene set associations was then taken to give the $p$ value for the gene set comparison.

Subsets of this dendrogram were then created by cutting the tree at the height which gives 15 unique groups. We tested these subsets for gene set enrichments with the DAVID online enrichment tool[53] using all genes in the individual annotation matrix as background.

**Gene-expression/environment comparison.** Sample meta data were downloaded from the CAMERA database as described above and included data on 12 measured variables. Additionally, using the downloaded CDS data for each sample, we calculated the genome-wide usage of codons encoding each amino acid.

Significance of expression/environment associations was calculated using the combined annotation matrix data and calculating the Spearman correlation between all the samples with both expression and environmental data. These correlations were converted to $p$ values by permuting the environmental data 10,000 times and calculating the number of permuted correlations with an absolute value greater than or equal to the observed correlation, divided by the number of permutations as described above. These permutation $p$ values that beat all permutations were then converted to $z$-score $p$ values as described above.

Expression correlation with UV radiation was measured by downloading the radiation data from NCEP and mapping the latitude values to the values for downward solar radiation flux in the winter at each latitude[54].

**Addition of genes to gene sets.** The correlation of a gene with a gene set was calculated by finding the median PEP correlation of the gene with all the genes in the gene set. The significance of this correlation was calculated by finding the median PEP correlation of the gene with 10,000 permuted gene sets, created as described above, and summing the number of permuted medians with a greater correlation and dividing by the total number of permutations.

To test for genome-wide association study (GWAS) hit enrichment, a list of GWAS SNPs with $p$ value less than 0.05 was downloaded from the Genome-Wide Repository of Associations Between SNPs and Phenotypes (GRASP) database[55]. This database was then culled for SNPs with a type II diabetes association and GWAS SNPs in genes were matched to genes in this data set using human gene IDs. Enrichment of GWAS hits in the list of genes added to a gene set was calculated by taking the top ten genes by PEP $p$ value (with secondary ordering by correlation) with the set and comparing these GWAS $p$ values to 10,000 random samplings of the same number of GWAS $p$ values, asking how often a set of $p$ values smaller than all of the observed $p$ values was found by chance. To test for a genome-wide trend, the list of genes added to a gene set was divided into 1000 gene bins ordered by the $p$ value of PEP association and correlation to calculate the percent of human genes in each bin with a GWAS $p$ value in the database.

The absolute value of the Pearson correlation between gene bin and percent GWAS gene was then compared to 10,000 random permutations of gene ordering.

**Code availability**. Scripts used in this analysis were written using the R programming language (3.0.0) and are available upon request.

## Data availability

All original MMETSP data are available at http://camera.crbs.ucsd.edu/mmetsp/index.php. Our processed expression matrix is available at http://mmetspdata.appspot.com.

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

## Acknowledgments

We would like to thank the members of the Fraser Lab and D. Petrov for helpful discussions and advice, S. Guida for assistance with the MMETSP data, and R. Kita for creating the data website. This work was funded by NIH grant 2R01GM097171-05A1.

## Author contributions

H.B.F. and T.M.M. designed the study. T.M.M. performed data analysis. H.B.F. and T.M.M. wrote the manuscript.

## Additional information

**Competing interests:** The authors declare no competing interests.

