## [Peer Review File · Nature Communications]

Reviewers' comments:

Reviewer #1 (Remarks to the Author):

The manuscript "Comparative expression profiling reveals widespread coordinated evolution of gene expression across eukaryotes" by Martin and Fraser describes a new method for the analysis of coordinated evolution in the context of a large set of expression data from single celled eukaryotic species collected in the Marine Microbial Eukaryotic Transcriptome Project. The authors introduce their method, phylogenetic expression profiling (PEP) and compare to the phylogenetic profiling (PP) method which has been previously used. PEP is conceptually similar to PP, but has not previously been feasible in part because enough data for a method like PEP has not previously existed.

Overall the manuscript is well written, the method is clearly described and is apparently useful. The authors have released a processed version of the data matrix to facilitate wider analysis in the community. As such this paper is something of a combined methods/resource paper. In this role, it is certainly valuable.

Although the volume metagenomic data generated by the MMETP project mean that it was likely natural that this method would be developed with eukaryotic microbes. However, I would anticipate applicability in a wider range of species as relevant data is generated.

The following comments provided to help increase the impact of the presentation and to ensure that the results are placed in appropriate context of the relevant literature in other species.

The example of tRNA ligase levels should be put into appropriate context in the discussion in light of results in other eukaryotes such as those from Kutter, et al (10.1101/gr.176784.114 and 10.1371/journal.pgen.1006024).

There is a problem with the references in the manuscript. Only 32 are listed in the reference list, but at least 49 are referred to in the text.

Figure 1B has very little information. The authors should consider a more informative figure possibly by incorporating some aspects of figure S1 or simply remove 1B.

I downloaded the data from <http://mmetspdata.appspot.com>. It is a 20mb file. There is little to no documentation about the contents of the file and this should be provided. The data should also be hosted either with supplementary information for the paper for the purposes of long term storage or put into an appropriate permanent repository for flexible data storage such as BioStudies.

Reviewer #2 (Remarks to the Author):

The authors present a very original and imaginative approach to measuring co-expression that they show to contains a strong "pathway" signal. In principle I would support publication in Nature communications, given the that issues below are handled.

The manuscript goes at great length to compare the results with PP data. I wonder why there is no comparison with within one species (e.g. yeast) co-expression data.

why were the individualy and jointly annotated samples analyzed separately? The authors assume to

much knowledge of the reader about this sequencing project.

I find the conclusions with respect to Alzheimer and the Golgi very unconvincing. Figure 3A contains no information, Figure 3B gives the impression as picking the highest correlation among a large set of correlations. This is of no added value to the manuscript. Similarly, a p value of 0.03 does not convince me that PEP can help identifying diabetes genes. I understand the wish to show the medical relevance of this technique, but a comparison with protein-protein interaction data, specifically ones that cannot be detected with PP would be more convincing.

Does UV radiation scale linearly with latitude? The authors should be able to find UV radiation data rather than using latitude.

editorial:

The method is quite distinct from measuring conserved co-expression, and I think that the sentence "since such conservation could result from a lack of change in expression" is rather misleading. Maybe "since such conservation does not depend on change in expression" would be better.

The manuscript mentions UniProt100, I take it the authors mean UniRef100?

"Up to five Swissprot ID annotations provided by MMETSP for each CDS were then culled for IDs that had a BLASTP alignment score of at least 80% the maximum alignment score for that CDS"
Given that alignment scores scale linearly with identity and the diversity of sequences in general, requiring 80% of the maximum score for a CDS appears very high to me. In the case I am missing something, please explain better.

The subsequent sentence, and specifically the part "and then across samples by ranking the UniRef100 IDs by alignment score and then creating an expression vector for an annotation by matching the unique top ranked annotations against the top ranked annotation for each CDS " has eluded me or quite some rereads (I think I do finally understand it now), also after the examining Figure S1. Maybe two extra sentences that explain what the actual problem is would help (I guess it is that some CDS do not share top ranked annotations, this is mentioned in the manuscript, but please repeat it in the methods)

On page 7 and on other places the authors use the term "coordinated evolution" I find this a bit confusing, as I assume they are referring to coordinated expression evolution (Which may not be correct either as we do not know whether expression was measured under the same conditions). I would prefer to include the term "expression" this term, as coordinated evolution sounds as if they are examining the coordinated sequence evolution.

Reviewer #3 (Remarks to the Author):

This study reports a new method to define clusters of coordinately expressed genes across widely divergent lineages of eukaryotes. It uses expression profiles and evolutionary conservation to pinpoint

groups of conserved genes with common patterns of expression, and provides examples showing that these groups may be coding for protein complexes or specific pathways. The idea seems interesting and can provide a useful complement to phylogenetic profiling, where gene sets are grouped based on evolutionary conservation only. However, the study mostly describes the methodological development, with application to a data set that is not deeply investigated, and is sometimes used to infer conclusions for which it was not designed. This paper may be greatly improved by detailing results with other data sets and investigating deeper the outputs to delineate the interests and limitations of the method, which are unclear in the present version.

One may think that this method can be applied to define coordinated modules in many divergent eukaryotes, but the authors used mapping against the UniProt100, which limits to highly conserved genes, leading to only 4,219 different genes studied. This discards a high number of interesting cases in the MMETSP, which contains distant eukaryotes with divergent gene repertoires.

More generally, the limitations of the MMETSP gene set are not acknowledged, although it is possible that some should be relevant to this study. For instance, the method looks at differences between gene expression profiles, but are the authors sure that each MMETSP sample is a simple profile and not a mixture of conditions, as frequently done for this type of data sets, which tries to maximize gene discovery? Is this relevant for the detection of clusters?

The chapter "Detecting coordinated evolution of gene sets" is sometimes unclear, as the authors start with a data set and then used a second apparently based on human and mouse transcriptomes. It is difficult to follow there what is validation of the method and new results.

Importantly, the use of the 5,914 gene sets from a previous PP study is not made for estimating the interest of PEP compared to other co-expression based clustering. It should have been interesting to compare the power and resolution of PEP in this data set with that from other algorithms creating clusters based on expression profiling across conditions rather than across evolution.

Page 7, it is unclear from where the 218,791 gene pairs came from.

The authors used a confusing naming for the pathways they described. It is clear that diseases associated with multi-tissues organisms, like Alzheimer or diabetes cannot apply to pathways uncovered in the MMETSP. It seems incorrect to use these, rather than the biochemical pathways underlying the disease states, to describe the findings.

There also seems to be a confusion regarding the use of environmental data to analyse associations with expression profiles. To my knowledge, the environmental description associated with the MMETSP samples correspond to the sites of isolation of the strains. They do not describe characteristics of the culture conditions, nor environmental niches, and therefore cannot be used the way the authors did in page 8.

Reviewer #1 (Remarks to the Author):

The manuscript "Comparative expression profiling reveals widespread coordinated evolution of gene expression across eukaryotes" by Martin and Fraser describes a new method for the analysis of coordinated evolution in the context of a large set of expression data from single celled eukaryotic species collected in the Marine Microbial Eukaryotic Transcriptome Project. The authors introduce their method, phylogenetic expression profiling (PEP) and compare to the phylogenetic profiling (PP) method which has been previously used. PEP is conceptually similar to PP, but has not previously been feasible in part because enough data for a method like PEP has not previously existed.

Overall the manuscript is well written, the method is clearly described and is apparently useful. The authors have released a processed version of the data matrix to facilitate wider analysis in the community. As such this paper is something of a combined methods/resource paper. In this role, it is certainly valuable.

We appreciate this assessment.

Although the volume metagenomic data generated by the MMETP project mean that it was likely natural that this method would be developed with eukaryotic microbes. However, I would anticipate applicability in a wider range of species as relevant data is generated.

The following comments provided to help increase the impact of the presentation and to ensure that the results are placed in appropriate context of the relevant literature in other species.

The example of tRNA ligase levels should be put into appropriate context in the discussion in light of results in other eukaryotes such as those from Kutter, et al (10.1101/gr.176784.114 and 10.1371/journal.pgen.1006024).

We thank the reviewer for pointing these out, and have added these references to our Discussion section on tRNA abundance and codon usage (pg 11).

There is a problem with the references in the manuscript. Only 32 are listed in the reference list, but at least 49 are referred to in the text.

We apologize for the missing references, which have been restored to the list.

Figure 1B has very little information. The authors should consider a more informative figure possibly by incorporating some aspects of figure S1 or simply remove 1B.

We have updated Fig 1B with additional information. Our goal was to provide a simplified workflow that would be understandable to a wide audience of biologists, while providing the complete details (which may not be as easily understood by non-specialists) in Supplemental Fig S1.

I downloaded the data from <http://mmetspdata.appspot.com>. It is a 20mb file. There is little to no documentation about the contents of the file and this should be provided. The data should also be hosted either with supplementary information for the paper for the purposes of long term storage or put into an appropriate permanent repository for flexible data storage such as BioStudies.

We have added more detailed row and column labels to this file, and also added it to the set of supplemental files (Table S1) hosted by the journal.

Reviewer #2 (Remarks to the Author):

The authors present a very original and imaginative approach to measuring co-expression that they show to contain a strong "pathway" signal. In principle I would support publication in Nature communications, given that the issues below are handled.

The manuscript goes at great length to compare the results with PP data. I wonder why there is no comparison with within one species (e.g. yeast) co-expression data.

*We appreciate this idea and have made an entirely new figure (Fig 3) to address it. We measured co-expression across two types of perturbations in the yeast *Saccharomyces cerevisiae*: environmental perturbations, and genetic perturbations (gene knockouts). We found that overall PEP was complementary to both of these methods, with little correlation in the strength of signal between PEP and the other methods. We have generated figure panels to illustrate both the large-scale patterns (Fig 3A-B) and two specific examples (Fig 3C-D). We think this has greatly improved the manuscript by showing how PEP is largely independent of within-species co-expression patterns.*

why were the individually and jointly annotated samples analyzed separately? The authors assume too much knowledge of the reader about this sequencing project.

We apologize for not explaining that the same RNA-seq samples were annotated using two different pipelines by the MMETSP Consortium. In order to avoid including the same samples twice in our expression matrix, we analyzed these separately. We have added a statement about this to the Methods (pg 13).

I find the conclusions with respect to Alzheimer and the Golgi very unconvincing. Figure 3A contains no information, Figure 3B gives the impression as picking the highest correlation among a

large set of correlations. This is of no added value to the manuscript. Similarly, a p value of 0.03 does not convince me that PEP can help identifying diabetes genes. I understand the wish to show the medical relevance of this technique, but a comparison with protein-protein interaction data, specifically ones that cannot be detected with PP would be more convincing.

Our goal was to illustrate how PEP could be used to generate hypotheses about connections between genes and pathways, rather than to prove such connections (which would require careful experimental validation). We agree that the examples we explored are speculative, and have therefore moved the original Fig 3 to the Supplement.

Does UV radiation scale linearly with latitude? The authors should be able to find UV radiation data rather than using latitude.

Yes, UV is mostly a function of latitude. When we substitute UV radiation values (averaged across many years) for latitude, a strong correlation is still seen (Spearman's $r = 0.47$). We have added the plot with UV radiation values to the Supplement as Figure S5.

The method is quite distinct from measuring conserved co-expression, and I think that the sentence "since such conservation could result from a lack of change in expression" is rather misleading. Maybe "since such conservation does not depend on change in expression" would be better.

We agree, and have made this change.

The manuscript mentions UniProt100, I take it the authors mean UniRef100?

Yes, thank you for catching this error. We have corrected this mistake.

"Up to five Swissprot ID annotations provided by MMETSP for each CDS were then culled for IDs that had a BLASTP alignment score of at least 80% the maximum alignment score for that CDS" Given that alignment scores scale linearly with identity and the diversity of sequences in general, requiring 80% of the maximum score for a CDS appears very high to me. In the case I am missing something, please explain better.

We based this cutoff on the observation that typically there is a steep dropoff of alignment scores after roughly this point. We have added an explanation of this to the Methods section (pg 13).

The subsequent sentence, and specifically the part "and then across samples by ranking the UniRef100 IDs by alignment score and then creating an expression vector for an annotation by matching the unique top ranked annotations against the top ranked annotation for each CDS" has eluded me or quite some rereads (I think I do finally understand it now), also after the examining Figure S1. Maybe two extra sentences that explain what the actual problem is would help (I guess it is that some CDS do not share top ranked annotations, this is mentioned in the manuscript, but please repeat it in the methods)

We agree this was confusingly written, and have now attempted to explain our motivation more clearly: "Iterative matching is required as some CDS do not share top ranked annotations, but do

share lower ranked annotations. This initial round of matching was then followed by matching successively lower ranked annotations of still unmatched normalized read counts until all read counts are combined into expression vectors. Note that only unmatched read counts are carried over into each new iteration. For details see Supplemental Fig. S1.” (pg 13)

On page 7 and on other places the authors use the term "coordinated evolution" I find this a bit confusing, as I assume they are referring to coordinated expression evolution (Which may not be correct either as we do not know whether expression was measured under the same conditions). I would prefer to include the term "expression" this term, as coordinated evolution sounds as if they are examining the coordinated sequence evolution.

We have updated this term to be “coordinated expression evolution” or “coordinated evolution of gene expression” throughout the paper.

Reviewer #3 (Remarks to the Author):

This study reports a new method to define clusters of coordinately expressed genes across widely divergent lineages of eukaryotes. It uses expression profiles and evolutionary conservation to pinpoint groups of conserved genes with common patterns of expression, and provides examples showing that these groups may be coding for protein complexes or specific pathways. The idea seems interesting and can provide a useful complement to phylogenetic profiling, where gene sets are grouped based on evolutionary conservation only. However, the study mostly describes the methodological development, with application to a data set that is not deeply investigated, and is sometimes used to infer conclusions for which it was not designed. This paper may be greatly improved by detailing results with other data sets and investigating deeper the outputs to delineate the interests and limitations of the method, which are unclear in the present version.

One may think that this method can be applied to define coordinated modules in many divergent eukaryotes, but the authors used mapping against the UniProt100, which limits to highly conserved genes, leading to only 4,219 different genes studied. This discards a high number of interesting cases in the MMETSP, which contains distant eukaryotes with divergent gene repertoires.

We agree that the PEP method we have developed is only applicable to highly conserved genes. This is one consequence of studying evolution across such vast timescales; drawing conclusions from the unique gene repertoires of each individual microbe studied would be quite difficult, since their functions are almost always unknown. We hope (and expect) that PEP will still be a useful method for asking questions about conserved genes, as we show with several examples. We also note that phylogenetic profiling, which has led to numerous discoveries made by many different labs, is also restricted to genes that are highly conserved across vast evolutionary distances (with the additional requirement that the informative genes must have been lost from multiple independent lineages, which restricts its potential set of informative genes even further).

More generally, the limitations of the MMETSP gene set are not acknowledged, although it is possible that some should be relevant to this study. For instance, the method looks at differences between gene expression profiles, but are the authors sure that each MMETSP sample is a simple profile and not a mixture of conditions, as frequently done for this type of data sets, which tries to

maximize gene discovery? Is this relevant for the detection of clusters?

We agree that this is an important point. Based on the sample annotations provided by the MMETSP Consortium, it appears that each profile is indeed a single condition. We have added this sentence to the Discussion to highlight potential limitations of the MMETSP data: “Although the MMETSP data represent the most phylogenetically diverse collection of gene expression profiles produced to date, there is likely much to gain from profiling additional species and culture conditions.” (pg 12)

The chapter “Detecting coordinated evolution of gene sets” is sometimes unclear, as the authors start with a data set and then used a second apparently based on human and mouse transcriptomes. It is difficult to follow there what is validation of the method and new results. Importantly, the use of the 5,914 gene sets from a previous PP study is not made for estimating the interest of PEP compared to other co-expression based clustering. It should have been interesting to compare the power and resolution of PEP in this data set with that from other algorithms creating clusters based on expression profiling across conditions rather than across evolution.

*We appreciate this idea, and agree that comparing PEP to co-expression across conditions is very informative. We have made an entirely new figure (Fig 3) to address this. We measured co-expression across two types of perturbations in the yeast *Saccharomyces cerevisiae*: environmental perturbations, and genetic perturbations (gene knockouts). We found that overall PEP was complementary to both of these methods, with little correlation in the strength of signal between PEP and the other methods. We have generated figure panels to illustrate both the large-scale patterns (Fig 3A-B) and two specific examples (Fig 3C-D). We think this has greatly improved the manuscript by showing how PEP is largely independent of within-species co-expression patterns.*

Page 7, it is unclear from where the 218,791 gene pairs came from.

We apologize for not explaining this. We have added an explanation to the Methods: “For comparing pairs of gene sets, we identified 218,791 gene set pairs that each had at least three genes in our combined expression matrix.” (pg 17)

The authors used a confusing naming for the pathways they described. It is clear that diseases associated with multi-tissues organisms, like Alzheimer or diabetes cannot apply to pathways uncovered in the MMETSP. It seems incorrect to use these, rather than the biochemical pathways underlying the disease states, to describe the findings.

We agree with this idea in principle, but in these two cases there is no underlying biochemical pathway that corresponds to these gene sets; rather, they are a collection of genes known to contribute to these diseases, which include many different underlying pathways. We have attempted to address this concern in the Discussion: “Although it may at first seem surprising that unicellular eukaryotes could shed light on complex diseases like Alzheimer’s and T2D, the fact that the genes are present throughout eukaryotes suggests that the underlying cellular functions are far more conserved than the specific human disease phenotypes—consistent with previous work, for example using yeast to study Parkinson’s disease⁴¹ and plants to study neural crest defects⁴².”

There also seems to be a confusion regarding the use of environmental data to analyse associations with expression profiles. To my knowledge, the environmental description associated with the MMETSP samples correspond to the sites of isolation of the strains. They do not describe characteristics of the culture conditions, nor environmental niches, and therefore cannot be used the way the authors did in page 8.

We agree that the environmental data (e.g. latitude used in Fig 4A) represents the isolation site of each strain, and is not related to the culture conditions. However, this was the goal of our analysis. We reasoned that if microbes have adapted to live at different latitudes, then we may see a signature of this by comparing the latitude at which they were isolated with their gene expression levels. This analysis therefore only works if a gene's expression level in culture reflects its adaptation to its natural environment, which should often be the cases if it is genetically encoded (as opposed to environmentally-induced). Indeed, we did find some evidence for this, with our top hit for association with latitude being a known UV-repair related protein, POLH. We have added a new supplemental figure (Figure S5) to demonstrate that the correlation is also highly significant when using estimated UV radiation at each isolation site, rather than latitude. We hope that as other large RNA-seq data sets of diverse species are collected in the future, similar methods might be applied to discover other local adaptations.

REVIEWERS' COMMENTS:

Reviewer #1 (Remarks to the Author):

All of my comments have been adequately addressed.

Reviewer #2 (Remarks to the Author):

I have no more issues with this manuscript. Nice and original work, please publish.

Reviewer #3 (Remarks to the Author):

This new version contains appropriate answers to most of my previous concerns, in particular there is a useful addition of comparisons between PEP and co-expression in single species. There is still one point that necessitates additional writing, that of detection of gene expression adaptation with latitude. The authors did not change their text there, but I still find that there is a risk of confusion. They only used latitude of isolation of the strain, which cannot be confounded with the breadth of latitudes where an organism lives, nor with the actual culture conditions. There are many ongoing studies of in situ transcriptomics with much more potential to reveal such correlations. The argument of the authors that if an organism was there it should be genetically adapted is not sufficient. Sentences have to be changed to explain exactly what the authors did, which is ambiguous in the current text.

Reviewer #3:

This new version contains appropriate answers to most of my previous concerns, in particular there is a useful addition of comparisons between PEP and co-expression in single species. There is still one point that necessitates additional writing, that of detection of gene expression adaptation with latitude. The authors did not change their text there, but I still find that there is a risk of confusion. They only used latitude of isolation of the strain, which cannot be confounded with the breadth of latitudes where an organism lives, nor with the actual culture conditions. There are many ongoing studies of in situ transcriptomics with much more potential to reveal such correlations. The argument of the authors that if an organism was there it should be genetically adapted is not sufficient. Sentences have to be changed to explain exactly what the authors did, which is ambiguous in the current text.

We have clarified in the Results section that our analysis involved the latitude where each sample was collected. In addition, we added this to the Discussion: “One caveat of this analysis is that it relies on the latitude of isolation of each strain and does not represent the breadth of latitudes where a species lives or the culture conditions. Other methods, such as directly measuring gene expression from environmental samples without culturing, could further elucidate these types of relationships.”